# Carbapenems as Antidotes for the Management of Acute Valproic Acid Poisoning

**DOI:** 10.3390/ph17020257

**Published:** 2024-02-17

**Authors:** Nataša Perković Vukčević, Vesna Mijatović Jovin, Gordana Vuković Ercegović, Marko Antunović, Igor Kelečević, Dejan Živanović, Slavica Vučinić

**Affiliations:** 1National Poison Control Centre, Military Medical Academy, 11000 Belgrade, Serbia; 2Medical Faculty Military Medical Academy, University of Defense, 11042 Belgrade, Serbia; 3Department of Pharmacology, Toxicology and Clinical Pharmacology, Faculty of Medicine, University of Novi Sad, 21000 Novi Sad, Serbia; 4Faculty of Medicine, University of Novi Sad, 21000 Novi Sad, Serbia; 5Department of Psychology, College of Social Work, 11000 Belgrade, Serbia

**Keywords:** poisoning, valproic acid, VPA, carbapenems, meropenem

## Abstract

Introduction: Valproic acid (VPA) is a broad-spectrum drug primarily used in the treatment of epilepsy and bipolar disorder. It is not an uncommon occurrence for VPA to cause intoxication. The established treatment of VPA poisoning includes supportive care, multiple doses of activated charcoal, levocarnitine and hemodialysis/hemoperfusion. There is a clinically significant interaction between carbapenem antibiotics and VPA. By affecting enterohepatic recirculation, carbapenems can increase the overall VPA clearance from the blood of intoxicated patients. It is suggested that carbapenems could successfully be used as antidotes in the treatment of acute VPA poisonings. The aim: To evaluate the effectiveness of carbapenems in the treatment of patients acutely poisoned by VPA. Patients and methods: This retrospective study included patients acutely poisoned by VPA and treated with carbapenems at the Department of Clinical Toxicology at the Military Medicinal Academy in Serbia for a two-year period. Results: After the admission, blood concentrations of VPA kept increasing, reaching their peak at 114–724 mg/L, while the mental state of the patients continued to decline, prompting a decision to introduce carbapenems. After the introduction of carbapenems, the concentrations of the drug dropped by 46–93.59% (average 72%) followed by rapid recovery of consciousness. Ten out of eleven patients had positive outcomes, while one patient died. The most commonly observed complication in our group of patients was bronchopneumonia. Conclusions: The application of carbapenems for the management of acute VPA poisoning might be a useful and effective treatment option.

## 1. Introduction

Valproic acid (VPA) is a broad-spectrum drug that is primarily indicated in the treatment of epilepsy, being used as a monotherapy or in combination with other antiepileptic drugs. It is also indicated in the management of bipolar disorder and as a prophylactic treatment for migraine headaches [1,2,3]. Sometimes, it can be used to treat chronic neuropathic pain and fibromyalgia [4]. The off-label use is related to the treatment of schizophrenia, borderline personality disorder and acquired brain injury [4,5].

Upon absorption from the gastrointestinal tract, a significant proportion of the VPA is bound to plasma proteins (90–95%), while the rest of the drug (6–10%) remains unbound, representing a free, pharmacologically active, fraction of the drug [6]. The metabolic sites of VPA are mainly liver cells, where the drug is predominantly altered by glucuronidation (50%), mitochondrial β-oxidation (40%) and cytosolic ω-oxidation (10%) [2,7,8]. From the liver, the metabolized glucuronide conjugate travels to the kidneys to be excreted via urine. Only 3% of the drug is excreted unchanged [1,2]. 

It is not an uncommon occurrence for VPA to cause intoxication. Such intoxications can happen deliberately by a patient or unintentionally, due to other reasons, when toxic blood levels can be reached at therapeutic doses [1,9]. The therapeutic range of VPA is from 50 to 100 mg/L. The toxic effects appear when serum levels exceed 100 mg/L [9]. Intoxicated patients usually present to the emergency departments with signs and symptoms of central nervous system and/or respiratory depression, sometimes accompanied by cardiopulmonary arrest and/or acute renal failure [10]. These patients often have mild to moderate CNS depression, showing signs of encephalopathy associated with dose-related hyperammonemia [1,9,11,12]. It is noteworthy to mention that an increased ammonia level can be found in about one third of the patients that use VPA and it is therefore recommended to check it in these patients [13]. These poisonings can sometimes be so severe as to cause a fatal outcome in some patients [14,15]. The established treatment of VPA poisoning includes supportive care, protection of the airway by endotracheal intubation, gastric lavage, multiple doses of activated charcoal, levocarnitine and hemoperfusion/hemodialysis [1,9,12]. Due to the risk factors, shortcomings and difficulties [1,9,12,16,17,18] associated with the traditional treatment of VPA poisoning, a novel approach to the treatment of such intoxication should be considered.

The clinically significant interaction between carbapenem antibiotics and VPA is well known and important to keep in mind in the clinical practice [1]. This interaction, whose mechanisms are still not fully understood, leads to the accelerated decline of the VPA concentration in the blood of intoxicated patients [1]. Relevant literature shows that meropenem can lead to a 50–80% decline in the mean plasma levels of VPA [1], and that meropenem and ertapenem are more effective than imipenem in bringing down VPA blood levels [19,20]. Such a decline in VPA levels usually occurs during the first 24 h following the administration of carbapenems [11]. Higher doses of carbapenems are not more effective in lowering concentrations of VPA [21]. VPA levels were shown to return to the reference range up to 7 days following the discontinuation of carbapenem therapy [12,22]. Despite the recommendation not to use these drugs concurrently if the patient has epilepsy [23,24], the aforementioned interaction still implies that carbapenems could successfully be used as antidotes in the treatment of VPA poisonings [1]. 

The aim of this study is to retrospectively evaluate the effectiveness of carbapenems in the treatment of patients acutely poisoned by VPA by analyzing their medical records in relation to previously published data.

## 2. Patients and Methods

This retrospective study including patients with acute VPA poisoning treated with carbapenems at the Clinic for Emergency and Clinical Toxicology of the National Poisons Control Center (NPCC) was conducted from January 2021 to June 2023. NPCC is a referent institution in Serbia (approximately 7 million inhabitants) that provides medical treatment services for acute poisoning as well as the detection of chemical substances in biological materials. The following cases outline the use of carbapenems as antidotes for 11 patients who presented with intentional overdoses of VPA. Ultra-performance liquid chromatography coupled with the mass spectrometer (UPLC-MS) method was used to detect and quantify VPA in patient plasma samples [25]. VPA plasma concentrations were measured at the admission and monitored regularly. The Poisoning Severity Score (PSS), a system that grades the severity of poisoning based on the overall clinical course, regardless of the ingested dose, concentration, type and number of agents involved, was used [26]. The information on ingested dose was reported autoanamnestically or heteroanamnestically.

This study was conducted in accordance with the Declaration of Helsinki and its amendments after obtaining approval from the Ethical Committee of the College of Social Work, Belgrade, Serbia (No. 7/9).

## 3. Results

Our study included 11 patients, aged from 22 to 68 years (mean 45.45 years, median 43 years). Four patients (36.36%) were male individuals, and seven of them (63.64%) were female. The patients’ age and gender, along with the ingested VPA dose, VPA concentrations (upon admission, maximal and after 24 h) and the percentage of VPA concentrations reduction can be observed in Table 1. The coingested substances and their reported dose and measured concentrations upon admission are described in Table 2. The main clinical findings and PSS (Poisons Severity Score) are summarized in Table 3. During the treatment, the patients were under electrocardiography (ECG) monitoring with supportive measures being applied along with intravenous (IV) fluids. Low-molecular weight heparin (LMWH) was used to prevent deep-vein thrombosis (DVT) and thromboembolic events. Depending on the clinical course of intoxication, other measures of treatment were employed. The selected treatment modalities (activated, charcoal, intubation, mechanical ventilation, vasopressor support and carbapenem of choice) can be found in Table 4. A decision to administer carbapenems was made because VPA plasma levels kept increasing in comparison with its levels determined upon admission and the mental state of the patients continued to decline. From the moment these observations were made and confirmed, carbapenems were initiated. Therefore, 10 patients (90.9%) received meropenem for such purposes and imipenem was given to one patient (9.1%). The standard dose of selected carbapenem (1g q8h intravenously) was administered to all patients. Twenty-four hours after the highest levels of VPA were measured, the concentrations of the drug dropped by 46–93.59% (average 72%) in our group of intoxicated individuals. The changes in VPA concentration over time for all patients can be seen in Figure 1. The outcome was positive in 10 out of 11 patients, while one patient died. 

**Patient 1** A 60-year old woman with a medical history of psychiatric disorders presented to the emergency department several hours after she intentionally ingested VPA (extended-release) and concomitant psychiatric medications (Table 1 and Table 2). Upon arrival she was with impaired consciousness (stupor) and stable vitals (TA 110/90 mmHg, HR 96/min, SatO_2_ 96%). The initial concentration of ingested drugs can be seen in Table 1 and Table 2. Within 8 h after admission, the patient’s GCS was 3 and the TA was 70/40 mmHg. The appropriate treatment in the intensive care unit was initiated (Table 3). As the VPA plasma concentration kept increasing (Table 1, Figure 1), meropenem was introduced. VPA concentrations dropped to therapeutic levels (below 100 mg/L) 12 h after meropenem introduction and rapid cardio-circulatory stabilization with consciousness recovery was registered. Rhabdomyolysis (creatine kinase (CK) 5425 U/L) and aspiration pneumonia were the main complications, and both were successfully treated. When she was toxicologically recovered, the psychiatry team offered her psychiatric hospitalization, medication management and safe discharge planning. 

**Patient 2** A 39-year old patient with a history of psychotic disorders was admitted to the Clinic for Emergency and Clinical Toxicology of the NPCC for acute severe intentional drug intoxication. According to the patient’s family history, he had taken a combination of psychotropic medications (Table 1 and Table 2) for suicidal purposes. This was the patient’s first suicide attempt. The drug intake occurred 2 h before the patient was transported to the NPCC. Upon admission to the ward, the patient was unconscious, areactive, with a heart rate of 120/min, a TA of 125/75 mmHg, a SatO_2_ of 95% and pinpoint pupils. Laboratory tests showed hyperglycaemia (24.5 mmol/L) and the presence of reportedly ingested drugs whose concentrations are available in Table 1 and Table 2. Immediately after the admission, the patient went into respiratory failure with severe hypotonia, so the appropriate treatment was initiated in a timely manner (Table 4). A dialysis line was inserted and this was followed by a hemoperfusion procedure. Hyperglycaemia was the reason for the insulin pump usage and refractory hypotension for intravenous lipid emulsion application. In order to reduce high VPA concentrations (Table 1, Figure 1), meropenem as an antidote was advised. Despite all the mentioned measures of intensive treatment, hypotension was persistent, and severe bradycardia and then asystolic cardiac arrest occurred. Resuscitation efforts were undertaken but a return of a hemodynamically efficient heart rate was not achieved. In view of the ineffectiveness of all the performed activities, the patient was pronounced dead.

**Patient 3** A previously healthy 61-year old woman was found unconscious with a Glasgow Coma Score of 6 and non-reactive miotic pupils. The patient’s husband gave information that he had found several empty boxes of her own psychiatric medicines (Table 1 and Table 2). An electrocardiogram revealed a sinus rhythm, an HR of 90/min and a long QT interval (QTc > 500 ms). She was hypotensive (TA 70/40 mmHg), with a SatO_2_ of 95%. Laboratory tests upon admission were within the reference range. Toxicological analysis of the plasma of the patient identified the ingested drugs and their concentrations (Table 1 and Table 2). The patient was referred to the intensive care unit of the toxicology ward for symptomatic and supportive treatment (Table 4). As the concentration of VPA was increasing over the first 8 h, specific antidotal treatment with meropenem was started on the second day of hospitalization and was followed by a rapid decline of VPA concentration (Table 1, Figure 1). Hemodynamic stabilization, renal function and diuresis recovery were registered after 48 h, while consciousness was restored 72 h after admission. She was extubated on the fourth day of hospitalization. Aspiration bronchopneumonia was successfully treated with dual antibiotic therapy. 

**Patient 4** A 22-year old man was admitted to the emergency department of the NPCC approximately 2 h after ingesting a couple of medications that belonged to his regular therapy (Table 1 and Table 2). Upon admission, the patient was in sopor, maintaining coordinated defensive reactions to rough stimuli. The pupils were with sluggish response to light. Physical examination findings were as follows: a temperature of 36.7 °C, a blood pressure of 155/86 mm Hg, a heart rate of 105 bpm, and a peripheral oxygen saturation of 96% (breathing room air). With alcohol being the main concomitant (Table 2), the initial plasma concentration of VPA kept rising during the first 8 h after the admission (Table 1, Figure 1). Therefore, meropenem was used according to the protocol until he regained consciousness—48 h after the admission. 

**Patient 5** The patient, a 58-year old woman, was found unconscious in her home. An assumption as to the ingested drugs was made based on the emptied medication boxes found by emergency medical services (Table 1 and Table 2). The patient was diagnosed with glioblastoma from 2021, which was surgically removed, and treated with radiation and chemotherapy. Laboratory testing confirmed a high level of VPA (Table 1). Even though she showed a favorable response to meropenem (improvement of mental status), on the fourth day of hospitalization her mental state declined once again. Regardless of this fact, the outcome of her toxicological treatment was positive (Table 1, Figure 1). A CT scan of the head confirmed the progression of malignant disease (terminal phase), and the patient’s family was encouraged to start with a palliative treatment. 

**Patient 6** A somnolent 53-year old patient was admitted to the NPCC after the ingestion of medications that belonged to his regular treatment of epilepsy (Table 1 and Table 2). Initial toxicological analyses showed the presence of VPA and diazepam, whose concentrations can be observed in Table 1 and Table 2. During the first 8 h of hospitalization, his level of consciousness was decreased to a coma with a GCS of 3, requiring appropriate supportive treatment (Table 4). After the introduction of meropenem, the concentration of VPA was dropped to within the therapeutic range (Figure 1) and the patient’s consciousness was restored. He was discharged from the hospital fully recovered. 

**Patient 7** A 27-year old female with a previous history of opioid addiction was brought to the NPCC by her boyfriend for evaluation and treatment of her abnormal behavior. She had ingested VPA that belonged to her boyfriend 2 h before her arrival to the ED (Table 1). Upon admission, she was in sopor, maintaining coordinated defensive reactions to rough stimuli with dilated pupils that were reactive to light. Considering her medical history, she suffered from deep vein thrombosis, taking rivaroxaban regularly. Her vitals were as follows: a TA of 125/75 mmHg, an HR of 105 bpm and a SatO_2_ of 88%. The measured concentrations of VPA can be seen in Table 1. Regarding the treatment, meropenem was used as a specific antidote successfully (Figure 1). She was discharged from the hospital 2 days after the admission with advice to visit a cardiologist due to persistent sinus bradycardia during the toxicological hospitalization. 

**Patient 8** A 27-year old female patient with a past medical history of polysubstance use disorder was brought by ambulance to the Department of Clinical Toxicology for intentional VPA overdose (Table 1). She was found unconscious, with pinpoint pupils, a SatO_2_ of 85%, and blood pressure that could not be measured. Prior to admission, naloxone, intravenous flumazenil and oxygen via the oropharyngeal airway were administered along with the initiation of vasopressor support (Table 4). Upon admission, she was in a coma, without defensive reactions to rough stimuli, a TA of 100/50 mmHg, an HR of 90 bpm and a SatO_2_ of 98%. The patient’s initial significant toxicological findings confirmed the presence of VPA in the patient’s blood (Table 1). Due to a high VPA plasma concentration and severe clinical presentation (Table 3), meropenem was used as an antidote, which resulted in a gradual decline in its blood concentration (Figure 1). The intubation of the patient was required in order to perform pulmonary hygiene (Table 4). On the sixth day her consciousness was fully recovered and she was extubated. The main complication was aspiration bronchopneumonia treated successfully with antibiotics. 

**Patient 9** A 42-year old man was admitted to the medical assessment unit with acute confusion, drowsiness and slower voluntary movements. He had a medical history of bronchial asthma, regularly using bronchodilators. He was an opioid addict as well. He coingested VPA and zolpidem (Table 1 and Table 2) 2 h before the admission to the ED. Upon examination, his HR was 77/min, his TA was 145/95 mmHg and his SatO_2_ was 97%. Toxicological analyses revealed the presence of the abovementioned drugs (Table 1 and Table 2) in the patient’s blood. A favorable response was registered after the meropenem application (Figure 1) and the patient was discharged from the hospital on the second day after admission. 

**Patient 10** A 43-year old woman was hospitalized in the ICU after she ingested a combination of two psychotropic medications (Table 1 and Table 2). The patient was found on the street, without consciousness, surrounded by numerous tablets. The reported doses and measured blood concentrations can be found in Table 1 and Table 2. The patient was presented to the ED in a coma, with bilateral pinpoint non-reactive pupils, justifying intubation (Table 4). Symptomatic and supportive treatment as well as the usage of meropenem normalized the abnormalities and allowed for the disappearance of symptoms (Table 3, Figure 1). Following extubation on the second day of hospitalization, she was transferred to a psychiatry clinic for further treatment.

**Patient 11** A 68-year old woman was found unresponsive in her home with several empty boxes of different medications (Table 1 and Table 2). In her past medical history, high blood pressure and cerebrovascular insult were noted. Upon admission, she was afebrile and unresponsive to voice but had normal respirations and withdrew all extremities in response to noxious stimuli. Her pupils were constricted and reacted sluggishly to light. The measured concentrations of ingested drugs are presented in Table 1 and Table 2. After the introduction of meropenem, the normalization of VPA concentration was determined (Figure 1) followed by the recovery of mental status on the third day of hospitalization. Except for aspiration bronchopneumonia, no other complications were developed.

## 4. Discussion

The NPCC of the Military Medical Academy is the only referent institution in Serbia that is equipped with a sophisticated analytical laboratory that is able to conduct toxicological-chemical analyses in order to obtain information about the ingested drugs and their quantity immediately after the patient admission. Control analyses are then performed to track changes in the relevant drugs’ levels as well as the effectiveness of the applied treatment.

Occasionally, patients intoxicated by VPA with or without the co-ingestion of other drugs present to the Clinic of Emergency Medicine and Clinical Toxicology of the NPCC. Following positive results from relevant literature that are related to the well-established interaction between carbapenems and VPA, we decided to administer carbapenem antibiotics to the patients in our study as antidotes for VPA toxicity. This was the first experience of healthcare professionals in Serbia with carbapenems that were applied as specific antidotes for the treatment of acute VPA poisoning. 

To this date, the exact mechanism of interaction between carbapenems and VPA remains not fully understood. The acceleration of glucuronidation and increased distribution into red blood cells via efflux inhibition are some of the postulated mechanisms [11,27,28]. However, the predominant one, among several theories, is an enzyme inhibition theory. It was shown that carbapenems can irreversibly inhibit acylpeptide hydrolase, an enzyme responsible for the hydrolysis of glucuronic acid and VPA conjugate, which results in decreased enterohepatic recirculation. This, in turn, leads to the accelerated decline of the VPA concentration in the blood of intoxicated patients [1]. An increased clearance of valproate glucuronide due to the interaction with meropenem, a carbapenem antibiotic, was shown in an in vivo study in dogs, where the plasma concentration of VPA declined more quickly when meropenem was administered in comparison to the clearance rate without the concomitant meropenem administration [29].

Our study included 11 patients treated with carbapenems for acute VPA toxicity. According to the relevant literature, it was shown that meropenem can more effectively bring down VPA levels than imipenem [20]. Khobrani et al. pointed out that meropenem can reduce the half-life of VPA by as much as 56%, implying that this treatment can indeed be beneficial in some patients [12]. The half-life of VPA is a particularly important variable to be considered when dealing with intoxications. When it comes to VPA poisoning, it was shown that, in such scenarios, the half-life of this medication can increase two- to three-fold, being greater than 30 h [30,31]. This implies that the toxic effect of VPA lasts longer in patients in whose treatment carbapenems were not included, thereby having a negative influence on the duration of treatment [32,33] and its outcomes, and primarily the extent of potential complications that could stem from prolonged toxicity [34,35]. Taking into consideration the abovementioned facts, our carbapenem of choice in ten out of eleven patients was meropenem. From a toxicological point of view, the treatment was a success in nine patients, since their mental state improved quickly after meropenem administration. One patient was given imipenem as a specific antidote and he also recovered successfully from the poisoning. One individual passed away as a result of polydrug ingestion. This patient had a serious poisoning accompanied by a coma and respiratory and cardiocirculatory failure, which was complicated by bilateral bronchopneumonia. An extracorporeal circulation procedure was also initiated due to high toxic concentrations of the ingested drugs. Despite the applied measures of intensive treatment, a positive therapeutic response did not occur and the patient was pronounced dead 32 h after being admitted into the clinic. 

The number of reports regarding usage of carbapenems other than meropenem in these poisonings is not large. Nonetheless, other carbapenems can also be successfully applied in the treatment of VPA poisoning. For instance, as shown in our research, imipenem could be a beneficial carbapenem of choice. However, it is advised not to consider this carbapenem if a patient is with a history of seizures, since imipenem is considered the most epileptogenic carbapenem [20]. A positive treatment outcome using ertapenem as an antidote in acute VPA intoxication was reported by Doad et al. A 29-year old male presented to the ED with a deteriorated mental status. Laboratory results revealed an elevated VPA level, so it was decided by clinicians to administer ertapenem (1gIV, unidose) due to its long half-life in comparison to other carbapenems. Along with the effect of additional therapy, including levocarnitine and activated charcoal, the patient regained consciousness and was discharged from the ICU on the third day [36]. 

VPA poisonings do not occur often if the patient is regularly controlled and VPA levels are carefully monitored and taken into account when setting up the right dose for the treated individual. If the concentration of VPA is within the therapeutic range, there appears to be a small risk of this drug’s toxicity. However, if VPA levels exceed this range, acute poisoning may happen [5]. 

The symptoms of poisoning depend on the severity of intoxication, which is, for clinical purposes, usually categorized using the Poisons Severity Score into (0) an absence of poisoning, (1) minor poisoning, (2) moderate poisoning, (3) severe or life-threatening poisoning and (4) fatal poisoning [26]. Commonly, symptoms of VPA poisoning include, as reported by Pozarowska et al., confusion, sedation, coma, muscle weakness, weakened reflexes or areflexia, miosis, respiratory distress, metabolic acidosis, hypoglycemia, cardiovascular disorders, hypotension and circulatory failure or shock [5]. The symptoms that were reported in our study are in accordance with the ones that were mentioned above and they were predominantly an altered mental status (from somnolence to coma), miosis, respiratory distress and hypotension. 

It can be observed from our results that the concentration of VPA increased in some patients after the first measurement (upon admission). This increase may be related to the initiation of carbapenem treatment, which is usually not administered right after the admission but rather after some time. However, the delayed peak of VPA concentration can also be explained by the variable absorption of the extended release form of VPA. A case that supports this theory was reported by Thomas et al., when a 42-year old woman ingested quetiapine and divalproex sodium (an extended release formulation of valproate) in a suicide attempt. It was found that VPA levels were 134 mg/L, so activated charcoal and 1 g of intravenous ertapenem were administered for the management of elevated VPA levels and possible pneumonia. The result of this treatment was a rapid decline in VPA concentration, but it increased again due to the variable peak of the extended release form of the drug. Afterwards, meropenem 2 g q8h was introduced, following a rapid decline back to therapeutic levels of VPA and even subtherapeutic levels after some time. She was successfully extubated and transferred out of the ICU and discharged on 18th day of hospitalization with a VPA treatment of 500 mg twice a day [1]. Similarly, most of the patients in our study reached higher VPA plasma concentrations during their hospital stay (accompanied by a deterioration of consciousness) in comparison with VPA levels determined upon admission (Table 1). This fact was one of the main reasons for the administration of carbapenems as antidotes.

A patient with a medical history of epilepsy treated for intentional VPA and diazepam overdose was also described in our investigation (Patient 6). The highest level of VPA measured in this patient was 114 mg/L. After some time, the patient fell into a coma, so meropenem was introduced and was administered for the next 48 h. A favorable response to the treatment was achieved and the patient was eventually discharged. Despite the observed positive outcome in this case, caution is required when treating patients with epilepsy due to the increased seizure risk that is associated not only with the proconvulsive effect of carbapenems, but also with the described pharmacokinetic interaction between VPA and carbapenems. There is always a risk that carbapenem administration can bring down concentrations of VPA to subtherapeutic levels, thus leaving these patients prone to seizures. This concern was confirmed in an adult study by Huang et al. that included 54 patients, where they showed that the seizure rate increased after three seizure-free months in this cohort after they were administered carbapenem antiobiotics [19]. Therefore, it is mandatory to reconsider the therapeutic approach when treating intoxicated patients with epilepsy. Substitution of VPA by another antiepileptic drug might be one of the possibilities. Cunningham et al. reported a case of intentional VPA poisoning of a 38-year old female with a history of epilepsy. Her VPA concentration was 210.7 mg/L and it increased up to 243.6 mg/L three hours later. She was administered activated charcoal, levocarnitine for hyperammonemia (56 mmol/L) and 1 g of meropenem intravenously. Fourteen hours after the administration of carbapenem, the VPA concentration dropped to 30.3 mg/L. During the treatment, a decision was made by a neurologist to replace her previous antiepileptic medicine with lacosamide, which was started two days after the admission. As a result, no seizure activity was observed during her stay at the hospital [11]. 

Our decision to administer carbapenems as specific antidotes for acute VPA self-intoxication was influenced by a number of beneficial aspects of this treatment course. First, we took into consideration carbapenems’ high availability and their relatively lower costs than those for the other treatment methods, such as extracorporeal circulation, which are also not available at some hospitals [12,14]. Second, there appears to be a small risk for carbapenems administration to cause antibiotic resistance or an allergic reaction, since the onset of their pharmacokinetic effect aimed at VPA is rapid, so a higher number of doses is rarely necessary due to such effectiveness [12]. The effectiveness of carbapenems was undoubtedly proven among our patients by a significant decline in VPA blood concentrations, averaging 72%, after 24 h of carbapenems administration (Table 1). The safety of carbapenems is also established by the fact that, except for VPA, drug–drug interactions are not expected [37,38,39]. This absence of major clinical interactions of carbapenems with other groups of drugs (such as benzodiazepines or antipsychotics) is important because it can be expected that a high percentage of VPA poisonings is actually combined drug intoxications, as can be seen in our research. Third, since aspiration bronchopneumonia was commonly observed in our group of patients, carbapenems were also employed for the causal treatment of this complication. Finally, it was shown that activated charcoal and carnitine treatment can sometimes be of little to no success, as was reported by Sanivarapu et al. In their paper, a 42-year old woman, diagnosed with bipolar disorder and Crohn’s disease and known for multiple suicide attempts, ingested an unspecified quantity of extended-release VPA. She was hypertensive and tachycardic and had to be intubated to protect the airway. Since her VPA and ammonia levels kept rising together with the deterioration of her clinical status, she was administered activated charcoal and carnitine. However, her condition did not improve. Therefore, a decision was made to introduce 1 g of meropenem every 8 h. As a result, VPA levels dropped significantly after the first dose and, after the third dose, the concentration of VPA was 11.5 mg/L. Ammonia levels also decreased to 41 μmol/L. The patient’s mental state improved and she was successfully extubated and eventually discharged home [16]. 

The abovementioned valuable features of this method of VPA poisoning treatment were the main reasons why we opted for such a course of action. In our group of patients, a therapeutic success was observed in ten out of eleven patients. Therefore, we can postulate that there might be a relation between carbapenem administration and the successful management of VPA intoxication. However, without analytical studies, we are not able, at this point, to determine if there exists a causal relationship between this type of treatment and the observed favorable outcomes [40,41]. 

There is a limitation of the study and it is related to the missing information concerning the ammonia levels of intoxicated patients in our reported cases. The importance of the detection of these values is due to the following facts, as reported by Pegg et al.: there exists a positive correlation between the dose of ingested VPA and the onset of hyperammonemia and also a significant positive correlation between levels of VPA and ammonia in serum. Additionally, elevated valproate levels are not the necessary condition for hyperammonemia, since it can occur within normal ranges of valproate in the blood [42]. Regarding the above-stated facts, it is our opinion that checking the levels of ammonia in patients who overdose with VPA should be an essential element of routine diagnostic procedures for these individuals, since ammonia is implicated in the mechanisms of the encephalopathic process. Unfortunately, the detection of ammonia at our clinic is not a routine part of the standard diagnostic algorithm, since reagents are not always available to be used. However, it is highly advisable that these measurements become incorporated into such an algorithm.

## 5. Conclusions

Intentional self-poisoning with VPA is often severe and can be fatal if not treated adequately and on time. In such clinical scenarios, it is worth to consider the use of carbapenem antibiotics, since intoxicated individuals were shown to regain consciousness more rapidly after their administration. The presented investigation illustrates our positive experience with carbapenems as specific antidotes to VPA overdose. The application of carbapenems for the management of acute VPA poisoning might be a useful and effective treatment option for such intoxications. Further analytical studies are necessary to quantify and justify this choice of treatment.

## Figures and Tables

**Figure 1 pharmaceuticals-17-00257-f001:**
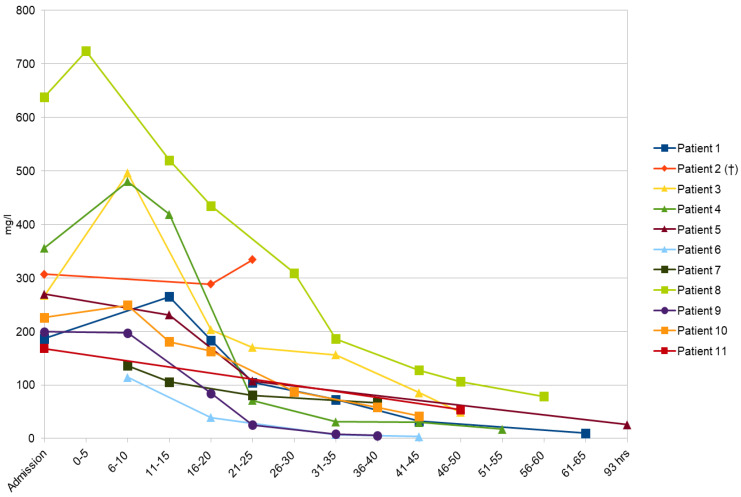
Changes in VPA plasma concentrations in intoxicated individuals over time. Legend: †—fatal outcome.

**Table 1 pharmaceuticals-17-00257-t001:** Summarized characteristics of intoxicated patients. Legend: Pt—patient; M—male; F—female; C_admission_—measured concentration of VPA upon admission; C_max_—the highest measured level of VPA; C_24h_—concentration of VPA measured 24 h after the highest measured level; NM—not measured; N/A—not applicable.

Pt	Age	Gender	Ingested Dose of Valproate (mg)	C_admission _ (mg/L)	C_max_ (mg/L)	C_24h _ (mg/L)	% of Reduction
1	60	F	30,000	186.80	265.00	52.15	80.32
2	39	M	unknown	306.76	344.18	NM	N/A
3	61	F	45,000	267.00	496.00	156.00	68.55
4	22	M	30,000	356.00	480.00	31.00	93.54
5	58	F	30,000	269.6	269.60	107.5	60.13
6	53	M	7,500	NM	114.00	7.31	93.59
7	27	F	30,000	NM	135.80	73.32	46.00
8	27	F	unknown	638.00	724.00	309.00	57.32
9	42	M	1,500	199.65	199.65	25.00	87.48
10	43	F	22,500	226.00	249.00	87.5	64.86
11	68	F	unknown	167.70	167.70	53.3	68.21

**Table 2 pharmaceuticals-17-00257-t002:** Coingested substances. Legend: Pt—patient; COD—Concentration upon admission; N/A—not applicable; LOD—limit of detection.

Pt	Concomitans	Dose (mg)	COD
1	fluphenazine	N/A	Below LOD
	bromazepam	180	0.27 mg/L
2	haloperidol	N/A	0.46 mg/L
	diazepam	N/A	1.03 mg/L
3	chlorpromazine	2500	1.68 mg/L
	lorazepam	50	0.19 mg/L
4	chlorpromazine	1250	0.26 mg/L
	alcohol	N/A	2.57‰
5	dexamethasone (N/A)	N/A	N/A *
6	diazepam (100 mg)	100	0.60 mg/L
7	/	/	/
8	/	/	/
9	zolpidem	200	0.2 mg/L
10	quetiapine	1500	3.16 mg/L
11	bromazepam	N/A	0.47 mg/L
	diazepam	N/A	0.27 mg/L
	lorazepam	N/A	0.26 mg/L
	enalapril	N/A	0.02 mg/L

* Measurement of concentration not performed due to technical reasons.

**Table 3 pharmaceuticals-17-00257-t003:** Clinical presentation. Legend: Pt—patient; PSS—poisons severity score; CNS—central nervous system.

Pt	PSS	Main Clinical Findings
1	3	Coma, hypotension, bronchopneumonia, rhabdomyolysis
2	4	Coma, hypotension, hyperglycaemia, respiratory failure, bronchopneumonia, asystole, death
3	3	Coma, bronchopneumonia
4	2	Sopor
5	3	Coma, bronchopneumonia, progression of CNS malignancy
6	3	Coma, bronchopneumonia
7	2	Sopor
8	3	Coma, hypotension, bronchopneumonia
9	1	Somnolence
10	3	Coma
11	3	Coma, bronchopneumonia

**Table 4 pharmaceuticals-17-00257-t004:** The selected treatment modalities. Legend: Pt—patient; I/MV—intubation/mechanical ventilation; N/A—not applicable.

Pt	Activated Charcoal	I/MV	Vasopressor Support	Carbapenem
1	Yes	Yes (I)	Yes	meropenem
2	Yes	Yes (I/MV)	Yes	meropenem
3	N/A	Yes (I)	Yes	meropenem
4	N/A	No	No	meropenem
5	N/A	No	No	meropenem
6	N/A	Yes (I)	No	meropenem
7	N/A	No	No	meropenem
8	N/A	Yes (I)	Yes	imipenem
9	N/A	No	No	meropenem
10	N/A	Yes (I)	No	meropenem
11	N/A	No	No	meropenem

## Data Availability

Data is contained within the article.

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
