# Peer review of "Carbapenems as Antidotes for the Management of Acute Valproic Acid Poisoning"

_pharmaceuticals, 2024, doi:10.3390/ph17020257_

Round 1

Reviewer 1 Report

Comments and Suggestions for Authors

this is a retrospective study, the author showed 11 case about Valproic Acid poisoning. actually, there are some studies about the retrospective study between the VPA and the Carbapenems. So this paper should be improved to attract the readers interest. there are some suggestion as following:

1. introduction part, the author should add references of other researchers reports about the VPA and the Carbapenems.

2. in discussion part,the author should add the mechanism of Valproic acid-carbapenem interaction

3. in the 11 cases, meropenem was used as an antidote for the poisoning of VPA. In clinical therapy,  are there other Carbapenems was usedas an antidote for the poisoning of VPA. 

Reviewer 2 Report

Comments and Suggestions for Authors

The present manuscript deals with carbapenem treatment of patients acutely poisoned with VPA. The study lasted two years, and sufficient data appears to be obtained. However, I have several questions and statements which I consider pertinent to be considered by the authors to produce a more consistent paper.

1)     How have the meropenem doses been adjusted in the different patients?

2)     After hospitalization, when did the patient start treatment with meropenem?

3)     The authors took into account that the toxic effects were due to the abusive use of VPA; however, other CNS substances were taken concomitantly with VPA by the patients. This topic needs to be discussed. What are the effects of meropenem on these substances (bromazepam, lorazepam, diazepam)?

4)     An Ethics approval should be added.

Reviewer 3 Report

Comments and Suggestions for Authors

I would like to thank the authors for an interesting manuscript. The topic is important as novel treatment strategies for intoxications often stem from case reports, and additionally, carbapenems are well known, simple to use, and have few immediate side effects (if ecological consequences of off-label wide spectrum antibiotics use are disregarded in this setting).

However, I think that some changes to the manuscript are required. 

Please shorten the introduction section so that it is to the point.

Please shorten the case report section - I suggest to aggregate patient data in text as most of the important information has already been added to the table. Please provide some additional basic information to the table as to the severity of presentation (I suggest adding the information on requirement of mechanical ventilation, vasopressor support and which substances were ingested in addition to VPA).

Please change the discussion section so as to provide comparison of your data to reports of VPA intoxication where carbapenems were not used, with focus on the differences in the clearance of VPA.

Please shorten the conclusion section to make it more to the point.    

Comments on the Quality of English Language

While the manuscript can be clearly understood, there are grammatical errors which need to be corrected. 

Round 2

Reviewer 2 Report

Comments and Suggestions for Authors

I agree with the authors answers and corrections . However, the ethics approval schould be updated such as: “This study was conducted in accordance with the Declaration of Helsinki and its amendments after obtaining approval from the Ethical Committee of the College of Socal Work, Belgrade, Serbia, (No. 7/9)”. Please Introduce this statement in an appropriate section: Ethics approval
